# Gut as an Alternative Entry Route for SARS-CoV-2: Current Evidence and Uncertainties of Productive Enteric Infection in COVID-19

**DOI:** 10.3390/jcm11195691

**Published:** 2022-09-26

**Authors:** Laure-Alix Clerbaux, Sally A. Mayasich, Amalia Muñoz, Helena Soares, Mauro Petrillo, Maria Cristina Albertini, Nicolas Lanthier, Lucia Grenga, Maria-Joao Amorim

**Affiliations:** 1European Commission, Joint Research Centre (JRC), 21027 Ispra, Italy; 2University of Wisconsin-Madison Aquatic Sciences Center at US EPA, Duluth, MN 55804, USA; 3European Commission, Joint Research Centre (JRC), 2440 Geel, Belgium; 4Laboratory of Human Immunobiology and Pathogenesis, iNOVA4Health, Faculdade de Ciências Médicas—Nova Medical School, Universidade Nova de Lisboa, 1099-085 Lisbon, Portugal; 5Seidor Italy Srl, 20129 Milano, Italy; 6Department of Biomolecular Sciences, University of Urbino Carlo Bo, 61029 Urbino, Italy; 7Laboratory of Hepatogastroenterology, Service d’Hépato-Gastroentérologie, Cliniques Universitaires Saint-Luc, UCLouvain, 1200 Brussels, Belgium; 8Département Médicaments et Technologies pour la Santé, Commissariat à l’Énergie Atomique et aux Énergies Alternatives (CEA), Institut National de Recherche pour l’Agriculture, l’Alimentation et l’Environnement (INRAE), Université Paris-Saclay, 91190 Paris, France; 9Instituto Gulbenkian de Ciência, 2780-156 Lisbon, Portugal; 10Católica Biomedical Research Centre, Católica Medical School, Universidade Católica Portuguesa, 1649-023 Lisbon, Portugal

**Keywords:** SARS-CoV-2 infection, COVID-19, gut microbiota, gastrointestinal disorders, enteric infection

## Abstract

The gut has been proposed as a potential alternative entry route for SARS-CoV-2. This was mainly based on the high levels of SARS-CoV-2 receptor expressed in the gastrointestinal (GI) tract, the observations of GI disorders (such as diarrhea) in some COVID-19 patients and the detection of SARS-CoV-2 RNA in feces. However, the underlying mechanisms remain poorly understood. It has been proposed that SARS-CoV-2 can productively infect enterocytes, damaging the intestinal barrier and contributing to inflammatory response, which might lead to GI manifestations, including diarrhea. Here, we report a methodological approach to assess the evidence supporting the sequence of events driving SARS-CoV-2 enteric infection up to gut adverse outcomes. Exploring evidence permits to highlight knowledge gaps and current inconsistencies in the literature and to guide further research. Based on the current insights on SARS-CoV-2 intestinal infection and transmission, we then discuss the potential implication on clinical practice, including on long COVID. A better understanding of the GI implication in COVID-19 is still needed to improve disease management and could help identify innovative therapies or preventive actions targeting the GI tract.

## 1. Introduction

While COVID-19 is mainly considered a respiratory disease, gastrointestinal (GI) symptomatology in COVID-19 has been reported, though the proportion varies depending on the studies, with patients reporting diarrhea, abdominal discomfort, nausea and/or vomiting, with diarrhea being the predominant GI symptom [1,2,3,4]. GI disorders, and particularly diarrhea, are proposed to be a direct consequence of SARS-CoV-2 intestinal infection, as many patients have detectable SARS-CoV-2 RNA in feces [5]. In addition, recent studies showed that non-human primates infected with SARS-CoV-2 had transient diarrhea [6]. However, the mechanisms leading to diarrhea in COVID-19 are largely unknown [7]. Studies from other viruses identified different mechanisms-inducing diarrhea such as malabsorption or inflammation secondary to enterocyte damage and death [8,9], the release of virulent toxins [8] and gut microbiota dysbiosis [10,11]. While SARS-CoV-2 RNA has been found in the stools of many patients, the impact of its presence in the GI tract remains to be clarified; in most cases, infectious particles are not recovered and infection does not always lead to diarrhea. In one study, 50% of the examined COVID-19 patients had a detectable level of virus in their feces, with only half showing diarrhea [12]. In another study the level of fecal viral load was positively associated with diarrhea [5]. Elevated fecal and serum levels of the inflammatory marker calprotectin in COVID-19 were not consistent with GI symptoms [13]. In line, limited intestinal inflammation was observed in patients with acute COVID-19 despite diarrhea, fecal viral RNA and SARS-CoV-2-specific immunoglobulin A (IgA) [14]. Thus, summarizing the current lines of evidence and uncertainties supporting intestinal infection and understanding the impact of intestinal SARS-CoV-2 on the GI system (epithelium damage, inflammation) could improve disease management, help to identify therapies or effective preventive actions targeting the GI tract.

Here, we used a methodological approach well-established in toxicology to evaluate key mechanisms driving SARS-CoV-2 mediated gut pathophysiology: the Adverse Outcome Pathway (AOP) framework which has been developed and is currently used to assess chemical risk for regulatory purposes. Based on existing data and available literature, the AOP approach seeks to pragmatically focus on essential biological key events (KE) at the different biological levels (molecular, cellular, tissue, organ, individual) up to an adverse outcome via a domino effect [15,16,17,18]. A KE describes a measurable and essential change in a biological system that can be quantified in experimental or clinical settings [19]. The strength of the relationship between the events is established by demonstrating biological plausibility and causality between pairs of events, called key event relationships (KER) [18,19]. The confidence that each KER occurs within an AOP is postulated by the evaluation of the weight of evidence [20]. Information contained in the KEs, KERs and AOPs are stored in an open access platform (https://aopwiki.org/) where they are identified by assigned unique numbers. Numbers in the text refer to these AOP-wiki pages. There, AOPs can be continuously updated as new information becomes available. This AOP approach highlights important inconsistencies and gaps in the evidence. Interestingly, based on a mechanistic understanding, AOPs help elucidate the pathophysiological mechanisms notably by learning from other inflammatory bowel diseases and other respiratory virus-related diseases (e.g., SARS, MERS, influenza) also presenting GI symptoms. This study was realized under the CIAO project which aims to make sense of the overwhelming flow of publications and data related to COVID-19 pathogenesis by using the AOP framework [21,22]. The project is based on the assumption that such mechanistic organization of the COVID-19 knowledge across the different biological levels will improve the interpretation and efficient application of the scientific understanding of COVID-19 [23]. In addition, we applied this methodology for the first time to map a viral disease of high societal relevance, expanding the AOP scope outside the toxicological field. 

We aim to evaluate if the gut can be an alternative route for viral entry, meaning that a productive intestinal infection by SARS-CoV-2 occurs and is responsible for the associated GI disorders (inflammation, permeability, diarrhea). To do so, we explored in the literature the evidence and uncertainties of each event starting with SARS-CoV-2 binding to its cellular receptor towards infection up to intestinal barrier disruption and intestinal inflammation. For this review, we collected evidence reported in tissue cultured cells, human samples and animal models of infection, including mice with human ACE2 (hACE2 mice), hamsters, minks, ferrets and non-human primates. 

## 2. Current Evidence and Uncertainties of an Active SARS-CoV-2 Enteric Infection

Besides GI symptoms experienced by many COVID-19 patients and SARS-CoV-2 RNA detected in feces, the rationale supporting intestinal infection was also based on the high level of expression in the intestines of the main SARS-CoV-2 cellular gateways: angiotensin converting enzyme 2 (ACE2) and transmembrane serine protease 2 (TMPRSS2). Enterocytes in the small intestine express the highest levels of ACE2 in the human body [24] and are one of the few human cell types that co-express TMPRSS2 [24], the main cofactor mediating cellular entry [7,25,26]. To evaluate if SARS-CoV-2 can effectively infect enterocytes, we assessed the evidence from the literature, starting with viable SARS-CoV-2 in the gut lumen binding to ACE2 receptors at the apical surface of the enterocytes, cell entry via TMPRSS2 cleavage and replication while antagonizing the antiviral response in order to release new viral particles (Figure 1). 

### 2.1. S Proteins Bind to ACE2 in Enterocytes and Mediates Viral Entry

*Biological plausibility.* Upon binding of SARS-CoV-2 to ACE2 (KE1739), the spike (S) proteins of the virus need to be activated through proteolytic cleavage to allow fusion between host and viral membranes, a key step in viral entry (KE1738), that releases viral RNA and proteins into host cells. Many proteases were identified as aiding in cell surface entry, such as TMPRSS2. Only three cell types showed co-expression of ACE2 and TMPRSS2, including enterocytes [26,27]. In addition, neuropilin-1 (NRP-1) was proposed to act as ACE2 co-receptor and promote, although to very low levels, SARS-CoV-2 entry even in cells that lack ACE2 and TMPRSS2 [24]. Maximum infection was reported when NRP-1 and ACE2 are co-expressed on the same cell types [28]. NRP-1 is reported to be expressed in the epithelia of the GI tract [29].

*Evidence*. Regarding ACE2 as the entry receptor for SARS-CoV-2, the level of ACE2 expression did not correlate with infectivity of cells in human intestinal organoids [30]. Both ACE2-positive and ACE2-negative SARS-CoV-2 infected cells in intestinal organoids were observed [31], potentially suggesting the existence of alternative entry receptors, ACE2 downregulation after infection, or reflecting expression levels under the detection limit. However, ACE2-knock-out (KO) intestinal organoids were fully resistant to SARS-CoV-2 infection [31], suggesting that ACE2 is the obligate entry receptor for SARS-CoV-2 in intestinal cells. Accordingly, in human gut-on-chip models composed of intestinal epithelial Caco-2 co-cultured with mucin-secreting HT-29 intestinal cells, the highest levels of ACE2 expression were found in the Caco-2 cells and after viral infection, Spike protein-positive Caco-2 cells were detected [32]. Similarly, higher ACE2 levels correlated with the maturity of enterocytes present in human differentiated enteroids and SARS-CoV-2 was able to infect ACE2+ mature enterocytes [33], therefore mature enterocytes are likely highly susceptible to infection. Further, the entry process was facilitated by TMPRSS2 and TMPRSS4 proteases [33]. However, when using CRISPR-Cas9 to generate different knock-out of key coronavirus host factors in human intestinal organoids, TMPRSS2, and not TMPRSS4, was found to be essential for SARS-CoV-2 entry [31].

As mouse ACE2 has a low affinity for S protein of SARS-CoV-2, different strategies were adopted to circumvent that mice are resistant to SARS-CoV-2 infection [34]. In a transgenic mouse expressing human ACE2 in the lungs, heart, kidneys and intestines (hACE2 mice), viral RNA was detected in the intestines of intranasally inoculated animals [35]. In another stable mouse model generated by CRISPR-Cas9 knock-in technology, hACE2 expression was also detected in the small intestine [36]. Another transgenic mouse model with hACE2 driven by the heterologous promoters (K18-hACE2) showed no viral RNA in the GI tract following nasal inoculation [37], but importantly hACE2 expression was not detected in the gut in these mice [37]. In Syrian golden hamsters, ACE2 protein is highly expressed in surface epithelium of ileum [38] and SARS-CoV-2 nucleocapsid protein was found in the intestine after intranasal infection [39]. In ferrets, ACE2 is expressed in the GI tract [40] and viral RNA was detected in the gut (ileum and colon) of intranasally infected male ferrets [39]. In rhesus monkeys, viral RNA was detectable in GI tissues after intranasal inoculation [41,42].

*Uncertainties, inconsistencies and gaps.* As previously noted, study in ACE2-KO intestinal organoids indicated ACE2 as the entry receptor of SARS-CoV-2 in enterocytes in vitro facilitated by TMPRSS2 [31]. The authors proposed that the discrepancy with the study considering TMPRSS4 could be explained by the expression in the KO organoids of physiological levels of the proteases rather than overexpression [31]. No studies have specifically investigated the role of NRP-1 in SARS-CoV-2 entry in the gut. However, it is interesting to note that different variants display different affinities for NRP-1, with omicron displaying higher affinity than previous variants. Future studies should elucidate whether this increase in affinity constitutes a functional evolutionary adaptation of SARS-CoV-2 to humans [43], and confer an advantage for viral entry. 

### 2.2. Viral Entry Leads to Antiviral Response 

*Biological plausibility*. Following cellular entry, the primary translation of the SARS-CoV-2 open reading frame (ORF) 1a and ORF1b genomic RNA produces non-structural proteins (NSPs) [44]. The ORF1a produces polypeptide 1a (pp1a) that is cleaved into NSP-1 through NSP11. A -1 ribosomal frameshift occurs immediately upstream of the ORF1a stop codon to allow translation through ORF1b, yielding pp1ab, which is cleaved into 15 NSPs (duplications of NSP1-11 and five additional proteins, NSP12-16). Viral proteases NSP3 and NSP5 cleave the polypeptides through domains functioning as a papain-like protease and a 3C-like protease, respectively [44]. The NSPs, structural proteins, and the accessory proteins are encoded by 10 ORFs in the SARS-CoV-2 RNA genome. They have multiple functions in evasion of the host innate immune response and in viral replication [45].

*Evidence*. The innate immunity activated by viral infections resulting in quick resolution of disease occurs in many instances of SARS-CoV-2 infection, such as in adults with no or mild symptoms [46], the young [47], and bats that harbor the virus without disease [48]. SARS-CoV-2 infection of human intestinal epithelial cells was associated with a robust innate immune response mediated by type III interferon, which inhibits SARS-CoV-2 replication and de novo virus production [49]. Interestingly, the scRNAseq study by Triana et al. found that SARS-CoV2 induced distinct proinflammatory and interferon-stimulated gene (ISG) expression profiles in infected and bystander cells in organoids. ISG expression was pronounced in bystander cells, while the infected cells showed strong NFkB/TNF-mediated pro-inflammatory response but a limited ISG expression. In intranasally infected hamsters, high levels of viral RNA were detected in the GI tract only in signal transducer and activator of transcription 2 (STAT2) KO animals suggesting that STAT2, the main actor of the interferon (IFN) response, is crucial for preventing intestinal virus replication and production of infectious progeny [50,51].

*Uncertainties, inconsistencies and gaps.* For SARS-CoV-2 infection, initial transcriptional analyses of infected cells have generated ambiguous results on the induction of type I/III IFNs and the subsequent expression of ISG and many studies associate better prognosis with increased innate immunity activation. However, the effectiveness of IFN treatment is still uncertain due to some studies evaluating IFN and other drugs [52]. There are uncertainties based on differing disease outcomes, mainly associated with the timing of administering IFN; administering late, in the inflammatory stage, led to long-lasting harm and worsened disease outcomes [52]. In the small intestine of infected hamsters, a mild antiviral gene signature was observed coinciding with a low-level inflammatory response and low replication similar to some human cases [53], in contrast to the robust replication seen in human small intestinal organoids [54] and severely ill patients [55].

### 2.3. Antagonized Antiviral Response Leads to Coronavirus Production

*Biological plausibility*. The SARS-CoV-2 virus has evolved a repertoire of proteins that bind and block proteins in the IFN cascade so the host antiviral proteins are not expressed, and the virus is free to replicate [56]. Interactions between SARS-CoV-2 proteins and human RNAs have been demonstrated to thwart the IFN response: NSP1 binds to 40S ribosomal RNA in the mRNA entry channel of the ribosome to inhibit host mRNA translation. NSP6 binds TANK binding kinase 1 (TBK1) to suppress interferon regulatory factor 3 (IRF3) phosphorylation, and NSP13 binds and blocks TBK1 phosphorylation [56]. NSP14 induces lysosomal degradation of type 1 IFN-alpha receptor (IFNAR) to prevent STAT activation [57]. ORF6 blocks nuclear import of IRF3 and STAT proteins to silence IFN-I gene expression [58]. ORF7a suppresses STAT2 phosphorylation and ORF7b suppresses STAT1 and STAT2 phosphorylation to block interferon-stimulated gene factor 3 (ISGF 3) complex formation with IRF9 [58]. ORF9b antagonizes IFN-I by targeting multiple components of RIG-I/MDA-5-MAVS, TOMM70, NEMO and cGAS-STING signaling [59,60,61,62]. The timely production of type I IFN by host cells is critical for limiting viral replication and promoting antiviral immunity [63]. If the antiviral response is antagonized (KE1901), the viral RNA can be translated, replicated, transcribed and the genomic RNA packaged before the new SARS-CoV-2 virions are assembled and released potentially into feces (KE1847). 

*Evidence*. In human intestinal organoids, following entry, gene expression analysis demonstrated that SARS-CoV-2 replicated with low induction of type I and III IFNs, though increased expression of ISG was observed [27]. Infection of Caco-2 cells leads to a weaker intrinsic immune response, associated with more de novo infectious virus production than T84 cells [49]. In ex vivo human intestinal tissues, SARS-CoV-2 replicated less efficiently (less viral genome copies produced, less infectious particles generated) but induced a more robust innate immune response than SARS-Co-V, including both type I and III IFNs while SARS-Co-V induced only IFNa expression [64]. These findings contrast with data obtained in ex vivo human lung tissues (SARS-CoV-2 replicated more efficiently while triggering an attenuated IFN response) [65]. Studies in human primary nasal epithelial cell cultures have shown that if exogenous IFN-I/III were administered intranasally prior to infection and at sufficient concentration, SARS-CoV-2 infection was inhibited [66]. Furthermore, in a hamster model IFN treatment limited tropism to distal tissues, including the intestine [53]. Also, some people have developed autoimmunity in which they produce autoantibodies that block IFN, resulting in more severe disease [67,68]. Loss of function variants in loci that control TLR3- and IRF7-dependent type I IFN immunity have been identified in a small number of severe adult patients with severe COVID-19 who had not been previously hospitalized for severe illness due to infection with other viruses [69]).

In humans, SARS-CoV-2 could productively replicate in surgically removed intestinal tissue but not in kidney or liver tissues [64]. SARS-CoV-2 RNA has been found in stools of infected individuals consistently, although with different frequencies (ranging from 15.3% to 81.8% of infected people [7]. In a retrospective cohort in China, the median duration of viral RNA in stool was 22 days [70]. In some patients, the viral load in feces reached 10^7^ copies/g suggesting an enteric infection not blunted by an interferon response [71]. In addition, SARS-CoV-2 RNA was reported to be detected in untreated wastewater sludge [72]. Viral RNA and intracellular staining of viral nucleocapsid protein were detected in GI epithelium from one patient in China who tested positive for SARS-CoV-2 RNA in feces [73] and duodenal biopsies of 2 out of 5 moderate COVID-19 patients; however, the staining was weak and scattered [74]. In another study, within six patients with GI symptoms subjected to endoscopy, SARS-CoV-2 RNA was detected in stomach, duodenum and rectum specimens of the two patients with severe disease, but only duodenum was positive in one of the four non-severe patients [75]. Another study that tested five COVID-19 patients, presenting with either upper abdominal pain or diarrhea. Early in infection, patients were subjected to a total of four esophagogastroduodenoscopy, and 2 in 5 showed signs of viral replication in the gut and increased numbers of antigen-experienced activated CD8^+^ T cells were detected within the epithelium [74]. This is in line with another study that found viral nucleocapsid in 5 out of 14 patients at an average of 4 months after initial COVID-19 diagnosis [76]. 

*Uncertainties, inconsistencies and gaps.* Contrasting results between ex vivo lung and intestinal tissues prove a line of evidence that SARS-CoV-2 infectivity and antiviral response is different in the gut than in lungs. Studies in lung cells and tissues showed that IFN expression is delayed or reduced by SARS-CoV-2 compared to influenza [66,77,78,79]. However, one exception to this observation is the response to high multiplicity of infection (MOI) response, where replication was robust with an observed IFN-I and -III signature. At low MOI, the virus might not be a strong inducer of the IFN-I and -III system, as opposed to conditions where the MOI is high [77]. hACE2 mice pre-treated with neutralizing antibodies against IFN-α/β receptors (mimicking pre-existing autoantibodies targeting type I IFNs) were more susceptible to SARS-CoV-2 infection with reduced survival [80]. Autoantibodies against IFN-α have been identified in patients with severe disease and have been shown to contribute to delayed viral clearance in lung cells [80]. 

While SARS-CoV-2 replication in human enterocytes in vitro is supported by strong evidence, evidence of SARS-CoV-2 infection in the digestive tract of animals showed mixed results. In the intranasally inoculated hACE2 transgenic mice, viral RNA was detected, but no infectious virus was isolated and no viral antigens were detected in the intestines [35]. In another stable mouse model generated by CRISPR-Cas9 knock-in technology [36], robust virus replication were demonstrated in lungs. When infected via the intragastric route, no data were reported on intestinal infection, but interestingly, these mice did exhibit lung infection [36]. K18-hACE2 showed no viral replication in the intestine following nasal inoculation [37]. This is coherent with the fact that hACE2 expression were not observed in the gut of the mice used in that study [37]. No studies in the gut have been reported so far for the HFH4-hACE2 mice developing severe/lethal disease. Thus, multiple strategies for introducing hACE2 into mice have been developed, a comprehensive characterization of the different models as well as of the doses and routes of inoculum used in each case is needed to correctly interpret the results [81]. In golden hamsters, expression of SARS-CoV-2 nucleocapsid protein was found in the intestine after intranasal infection [82]. In ferrets, viral RNA and viral subgenomic mRNA, indicative of a previous or current viral transcription, was detected in the gut but interestingly, in this case did not produce detectable infectious viral particles [39]. Viral RNA but no infectious virus was detected in the ileum of one over three minks inoculated intranasally [83]. In rhesus monkeys, viral RNA was detectable in digestive tissues and in fecal samples after intranasal inoculation and Tissue Culture Infective Dose (TCID50) assays suggested that the virus was infectious [41]. While in rhesus macaques infected via a combination of intranasal, intratracheal and ocular inoculation, viral RNA was and SARS-CoV-2 antigen were detected in the GI tract but not viral mRNA [42]. Thus viral RNA was detected in the intestines after virus inoculation (intranasal or intragastric) in almost all animal models [35,39,41,42,82] providing evidence for SARS-CoV-2 entry into enterocytes. However, evidence that the virus found in the GI tissues was infectious was observed only in rhesus monkeys in one study [41]. As already mentioned, these data calls for precaution of which models are suitable to study SARS-CoV-2 intestinal infection as well as for considering with care the doses and routes of inoculum used in each case [81]. Assessing intestinal infection and IFN response following infection with different dosages in the gut of all types of hACE2 infected mice and non-human infected primates [6] in parallel with ACE2 staining would help provide clear evidence of whether increased coronavirus production occurs in the gut in vivo. 

In humans, according to the few endoscopic and histological examinations based on one or two cases [73,74,75,76], the GI epithelium is potentially susceptible to infection by SARS-CoV-2 but to date, it remains unclear whether SARS-CoV-2 replicates in the human gut, and for how long it could persist in the gut. Even if difficult to obtain, further staining of COVID-19 patients GI epithelia, as well as omics analysis of intestinal biopsies notably regarding IFN response, are needed to confirm and quantify the proportion of patients with active replication in the gut.

## 3. Current Evidence and Uncertainties of SARS-CoV-2 Damaging Intestinal Barrier

SARS-CoV-2 infection was reported to be a cytopathic virus in lung cells triggering cell apoptosis in lung epithelial cells and in lungs of infected mice while lung sections of fatal COVID-19 patients revealed cell death markers [84]. Thus, SARS-CoV-2 has been proposed to induce cell death resulting in disruption of the epithelial monolayer integrity or alterations to tight junctions (TJ), the mucus layer and/or the cellular immune system. Disruption of the intestinal barrier layers is associated with increased intestinal permeability, also called “leaky gut”, which allows the transfer of commensal or pathogenic bacteria and bacterial components into the lamina propria and later on into the systemic circulation [85] (Figure 2).

### SARS-CoV-2 Production Impairs Intestinal Barrier

*Biological plausibility*. Within the host cell, the new virions are assembled (KE1847) and to release viral particles, the virus promotes host lysis, leading to cell death and compromising the integrity of the epithelial monolayer. The intestinal barrier is ensured by the integrity of the monolayer epithelium (via cell integrity and tight junctions/adherens proteins), together with the chemical barrier, the mucosal layer and the cellular immune system located in the lamina propria (KE1931). Alternatively, TJs might be altered following SARS-CoV-2 infection enhancing paracellular permeability. In addition, the mucus layer and/or the cellular immune system might be perturbed.

*Evidence*. No extensive cell death was observed after SARS-CoV-2 infection in intestinal organoids, compared to MERS-CoV that killed most cells within 48 h of infection [31]. SARS-CoV-2 also replicated less efficiently than SARS-CoV and induced less cytopathology in ex vivo human intestinal epithelium [64]. In contrast, studies in vitro with gut derived organoids report observable organoid disintegration [86] also associated with markers of apoptosis, such as caspase 3 [54,86]. No substantial histopathological changes were observed in the intestines of hACE2 intranasally inoculated mice in which no virus was isolated, nor viral antigens detected [35]. In Syrian hamsters, the histological analysis did not reveal intestinal damage or structural remodeling of the epithelium in hamsters but a trend towards increased blood concentration of intestinal fatty-acid binding protein (FABP), a systemic marker associated with disrupted gut integrity, has been detected [87]. This agrees with other studies [82,88] but contrasts with another study [89] in which severe epithelial cell necrosis and damaged intestinal villi were observed at 4dpi (but not at 2dpi). The reason for this discrepancy is unclear according to the authors, but they proposed differences in the virus preparations and the dose used to inoculate animals, which are well summarized in [81]. In rhesus monkeys, exfoliation of mucosal epithelium in the GI tract was observed after intranasal inoculation of SARS-CoV-2 as well as a reduced number of mucin-containing goblet cells at the earlier stage of infection [41]. In humans, no relation was noted between fecal calprotectin (FC) levels and fecal SARS-CoV-2 RNA in a cohort of 40 hospitalized patients with COVID-19 [1]. In another exploratory study, COVID-19 patients had elevated plasma levels of LPS-binding protein (a gut leakage marker) but not of intestinal FABP (a marker of enterocyte damage) [90]. These data suggest impaired gut barrier function without excessive enterocyte damage and highlight gaps to comprehensively understand under which experimental or clinical conditions, SARS-CoV-2 productively infects and kills enterocytes. However, in a human gut-on-chip model composed of intestinal epithelial Caco-2 co-cultured with intestinal mucin-secreting HT-29 cells, after SARS-CoV-2 infection, S-positive epithelial cells were detected along with damage to the intestinal villus-like structures, disturbance of the mucus layer and reduced expression of TJ (E-cadherin) [32]. Severe COVID-19 was associated with high levels of markers of tight junction permeability and microbial translocation [1,91,92], signaling a loss of the intestinal barrier function.

*Uncertainties, inconsistencies and gaps.* While the biological plausibility was high, currently, there is not enough evidence to support that enterocyte massive cell death following SARS-CoV-2 infection occurs systematically [93]. Number of cases showing histomorphologic changes due intestinal infection by SARS-CoV-2 is still limited. While not easy to obtain, more (post-mortem) intestinal biopsies of COVID-19 patients showing the presence of replicating SARS-CoV-2 along with cell death markers in epithelial cells of the small intestine would be needed to determine precisely if cell death occurs. A small body of evidence points toward a potential alteration of TJs upon SARS-CoV-2 infection. However, definitive evidence is still limited and warrants further research. The role of Ca/Zn/VitD depletions in COVID-19 patients which weaken physical tissue barrier integrity by interacting with TJ [94] is still unclear. Using biomimetic human intestinal gut-on-chip able to partially mirror intestinal barrier injury and response to viral infection [32] or human intestinal organoids could provide insight into the essentiality of these events in COVID-19. In addition, it would informative to assess the tight junction permeability in infected mice, hamsters or nonhuman primate models described above.

## 4. Current Evidence and Uncertainties of SARS-CoV-2 Enteric Infection Contributing to the Inflammatory Response

While productive replication still needs further studies, strong evidence supports SARS-CoV-2 entry in intestinal epithelial cells. There it might trigger a coordinated innate immune response due to the recognition of SARS-CoV-2 associated molecular patterns, similar to that reported in the lung cells [49,95], inducing an antiviral response as described above but also releasing proinflammatory mediators which recruit immune cells to the gut, which in turn secrete cytokines leading to gut inflammation (Figure 3).

### 4.1. Viral Entry Induces Pro-Inflammatory Mediators Release

*Biological plausibility*. Viral infections induce a proinflammatory response including expression of cytokines and chemokines via signal transduction pathways activation, such as NF-kB [96], JAK-STAT [97] and NFAT [98].

*Evidence.* Infection of human intestinal organoids with SARS-CoV-2 elicited a broad signature of cytokines [54] mediated by NFkB/TNF [30]. Lamers et al. [54] showed that the infection of human intestinal organoids with SARS-CoV-2 can induce Il7 expression. Intestinal viral infections cause IL22 expression in T cells via IFNβ1-mediated IL7 production by epithelial cells and IL6 production in fibroblasts. In non-human primates infected with SARS-CoV-2, increased serum concentrations of interleukin (IL)-8, IL-1RA, C-C motif chemokine ligand (CCL)2, CCL11, and chemokine (C-X-C motif) ligand (CXCL)13 were observed [6,42,99,100]. Higher levels of the pro-inflammatory cytokine IL-8 and lower levels of the anti-inflammatory cytokine IL-10 were detected in the feces of COVID-19 patients when compared to uninfected controls [14]. However, the lack of increase of other cytokines and of calprotectin in this study suggests that the immune response within the gut to this viral infection is limited.

*Inconsistencies, uncertainties and gaps*. Several components of inflammation exist but we have limited knowledge on the nature of inflammatory pathways triggered in the GI tract by SARS-CoV-2. Additional investigations in COVID-19 patients are still needed, such as analysis of in situ produced cytokines in gut biopsies from COVID-19 patients with distinct disease severity profiles. Of key importance it would be to dissect, if similarly to other enteric viral infections, to what extent does intestinal inflammatory response contribute to the systemic cytokine profile and which are the parallels and differences between in the epithelial response in the gut versus in the lung. In lung samples, a signature of low IFN-I and -III and high pro-inflammatory mediators was consistently observed in vitro, ex vivo, in vivo in longitudinal studies and in COVID-19 patients [77], performing similar analysis in the gut would be informative.

### 4.2. Pro-Inflammatory Mediators Recruit Inflammatory Cells in the Gut

*Biological plausibility*. Pro-inflammatory signaling (KE1496) recruits’ pro-inflammatory cells, such as neutrophils, macrophages, and T cells to the site of infection (KE1497).

*Evidence*. When cytokines are released, immune cells, such as neutrophils, macrophages, and lymphocytes, are recruited to the gut environment and facilitates an adaptive immune response [101]. In golden Syrian hamsters intranasally infected with SARS-CoV-2, the viral N protein was detected in the intestine, IL-4, IL-6, TNF-α and IL-12 were upregulated and the lamina propria exhibited mononuclear cell infiltration at 2dpi [102]. Histological examination of human intestinal samples revealed that lymphocytes and inflammatory cells infiltrate the lamina propria [103]. Neutrophils recruitment has been demonstrated by gut calprotein (neutrophil-specific alarmin protein) presence in COVID-19 patients where elevated fecal calprotectin and systemic IL-6 response were identified [1] and associated to intestine inflammation, adding to the evidence that SARS-CoV-2 triggers an inflammatory response in the intestine [104]. Recent studies also described that the cytokine storm may be associated with the expression of calprotectin [105,106] but another preprint study showed that the level of calprotectin was not linked to COVID-19 severity [14].

*Uncertainties*. Whether direct or indirect modulation in the gut immune activation during SARS-CoV-2 infection is responsible for immune cell recruitment needs to be examined more thoroughly. Direct cell death in intestinal epithelial and goblet cells can cause apoptosis and recruit immune cells or, alternatively, an indirect loss of GI tract integrity caused by viral infection can activate immune cell recruitment. A recent study in ferrets infected with SARS-CoV-2 by gavage compared the immunomodulation of probiotics in the duodenum, however a noninfected placebo group was missing, which would be informative [40]. Recently, in a non-human primate (rhesus monkey) model of SARS-CoV-2 infection, in vivo infection of GI tract increased apoptosis of intestinal epithelial and goblet cells along with intestinal inflammation by macrophages has been reported [41]. However, these results could not explain whe, ther immune modulation in the GI tract was due to direct infection of GI tract cells by the virus or due to changes in the GI tract integrity and microbiota under the influence of systemic cytokines and hypoxic conditions or a combination of all [104]. Certain patient subgroups such as the elderly and patients with type 2 diabetes or obesity have been shown to be associated with more severe disease [107]. Immune defense mechanisms at the digestive level are described as impaired in these populations [108,109]. It is not currently known whether this impaired digestive immunity is a risk factor for infection.

## 5. Current Insights, Research Needs and Potential Impact on Clinical Practices

### 5.1. Productive Enteric Infection

An important aspect is that to infect intestinal cells, viable SARS-CoV-2 must reach the gut lumen as an infectious particle, meaning able to actively replicate in the GI tract. In contrast to enteric viruses, enveloped respiratory viruses, such as influenza virus or SARS-CoV-2, are thought to be cleared by the exposure to digestive juices (gastric acid, bile, pancreatic juice) and mucus layer in the GI tract. SARS-CoV-2 was found to be extremely stable over a wide range of pH values (pH 3–10) [110] but rapidly lost infectivity in vitro in the low pH simulated gastric fluid (pH 1.5–3.5, fasting state) [33]. Furthermore, SARS-CoV-2 was rapidly inactivated in the lumen of the colon by enteric fluid [33]. This suggests that, predominantly, noninfectious particles reach the gut lumen [33]. However, while MERS-CoV also rapidly lost infectivity in fasting-state simulated gastric fluid, the infectivity was unaffected in fed-state-simulated gastric fluid (pH 5) [111,112]. Using human coronavirus OC43 (causing mild symptoms and not requiring a biosafety level 3 lab) as a surrogate for the pathogenic SARS-CoV-2, a very recent study showed that, except for fasting-state gastric fluid (pH 1.6), the virus remained infectious in all other GI fluids for 1 h and the presence of food improved viral survival in gastric fluids [112]. A similar strategy should be done for SARS-CoV-2 and investigate infectivity in this fluid simulating stomach acidity after meals. This would allow determining whether SARS-CoV-2 tolerates gastric acid and survives passage to the gut in all settings and whether SARS-CoV-2 ingestion with food could protect the virus against inactivation by the GI fluids [7]. Interestingly, in this regard, it was described that the usage of H pump inhibitors was associated with worse clinical outcomes for COVID-19 patients, despite not being associated with increased susceptibility to SARS-CoV-2 infection. This observation raises the question of whether some medicines permit SARS-CoV-2 replication in the gut [113]. In addition, if other conditions, including for example highly viscous mucus, protect virus particles, allowing the virus to retain its infectivity, as shown for influenza virions, should be evaluated to determine conditions in which SARS-CoV-2 could actively replicate in the gut [114]. In infected hamsters, SARS-CoV-2 intranasal infection was more efficient than oral infection. However, increasing viral dose in the initial inoculum, both intranasal and oral, resulted in higher levels of SARS-CoV-2 RNA in the lungs and in the intestines of these animals, suggesting that the initial dose is an important factor when considering gut infection and mechanisms that protect the virus from the harsh environment of the stomach [88]. Alternatively, other cell types may be able to transport SARS-CoV-2 to the gut, as for example, a small number of lymphocytes has been shown to be infected by SARS-CoV-2 [103] or even bacteria. A recent study showed that SARS-CoV-2 replicates outside the human body in vitro in bacterial growth medium, following bacterial growth and influenced by antibiotics administration, suggesting a bacteriophage-like behavior for SARS-CoV-2 [115] or the activation of other bacteriophages [116]. Electron and fluorescence microscopy images showed the presence of SARS-CoV-2 both outside and inside bacteria [116,117]. Further research is needed as these results could lead to a rethinking of SARS-CoV-2 biology and of effective management of COVID-19 transmission [115]. In addition, it cannot be excluded that both the viscous mucus and the gut microbiome could protect viral RNA and virus particles, allowing the virus to retain its infectivity. 

Thus, while the human gut expresses high levels of ACE2, and SARS-CoV-2 infection of human enterocytes in vitro is supported by strong evidence, human healthy gut may not be systematically permeable to viral entry due to the GI fluids, antiviral response and/or the protective multi-layers of the intestinal barrier. However, evidence of intestinal infection of SARS-CoV-2 has been reported, suggesting that there are some conditions that may render people susceptible to SARS-CoV-2 infection in the gut or that may protect the virus from degradation. For example, individuals with altered intestinal barrier prior to infection, or under certain medication or comorbidities, might be more vulnerable to gastrointestinal SARS-CoV-2 infection [118]. An inflammatory environment, as seen in many other conditions such as diabetes, obesity, or resulting from the cytokine storm in severe COVID-19, disrupting the intestinal barrier, may render the GI entry of the SARS-CoV-2 significant [33]. Different experimental models mimicking diseases known to be associated with an altered intestinal barrier exist. Literature describing their use to unravel the mechanisms behind SARS-CoV-2 GI infection is starting to emerge. A mouse preclinical T2DM/obesity co-morbidity model of COVID-19 [119] and a mouse model mimicking obesity-associated COVID-19 comorbidities were established [119]. Such models could accelerate the development of therapeutics for this highly susceptible population. Sex and diet-specific responses partially explaining the effects of obesity and diabetes on COVID-19 disease were observed [119]. The detrimental impact of continuous Western diet on COVID-19 outcome has been reported in Syrian hamsters [120]. The age dependent increase in disease can be observed in Syrian hamsters and nonhuman primates [81]. Thus age, medication, metabolic syndrome, via high fat diet for example, could be incorporated into models to mimic human comorbidities in order to investigate this important question. 

In addition, the colonic mucus barrier is shaped by the composition of the gut microbiota [121]. Alteration of the gut microbiota has been associated with severity in COVID-19 [122,123] and might contribute to disrupting the mucus barrier, rendering the gut more permissive to SARS-CoV-2. A body of evidence indicates that gut dysbiosis, prior to infection, represents a risk factor, meaning contributes to more severe outcomes in COVID-19 patients, potentially by modulating intestinal ACE2 expression, intestinal and systemic inflammation and gut barrier integrity [124]. 

In conclusion, further research is needed to acquire a comprehensive understanding of the conditions under which SARS-CoV-2 productively infects enterocytes in humans in vivo. Notably, it is important to understand if specific conditions, including age, comorbidities or medication are associated with release of infectious particles from feces by tracking and surveillance of several groups in the population. These studies could be complemented by in situ hybridization or staining of human tissues acquired from biopsies or post-mortem samples of gut retried from COVID-19 positive people.

### 5.2. Infectious Virus in the Feces

If SARS-CoV-2 can establish an intestinal infection, then it remains unknown whether infectious viral particles can tolerate GI fluids and be shed alive through feces with sufficient concentration and infectivity for subsequent transmission. Despite SARS-CoV-2 RNA being detected in stools, and the persistent viral shedding of SARS-CoV-2 in feces, current data from different studies are conflicting regarding the detection of infectious particles in feces. Infectious viral particles may be retrieved from anecdotal cases, although studies indicate that the vast majority of individuals infected with SARS-CoV-2 do not release infectious particles from stools [125]. While high viral RNA concentrations were observed in stools in two different studies (9 and 10 patients, respectively), infectious virus was not recovered in those samples [33,125,126]. In contrast, replicating SARS-CoV-2 virus was detected in feces in [127] and viable SARS-CoV-2 particles in stool samples in [128]. Several aspects could complicate SARS-CoV-2 isolation from fecal material, such as the stability of the virus in the feces [129] and the potential presence of numerous other viruses. These aspects could also make viral activity assays technically challenging. A procedure using filtered diluted specimens without the addition of any potentially toxic antibacterial agents and cell culture medium changing after centrifugation was described as responsible for high virus recovery [130]. Using a similar method Jeong et al. [128] failed to demonstrate the presence of viable virus in stools, but they were able to isolate SARS-CoV-2 from ferrets that were inoculated with stool samples from COVID-19 patients. Other SARS-like viruses have been isolated from animal feces (see [131] as example), coronaviruses related to SARS-CoV-2 were isolated from bat rectal swabs and guano and were tested as able to infect in vitro human cells [132]. Understanding whether and when fecal-oral transmission of SARS-CoV-2 might occur will be of critical importance for health workers since feces from infected hosts could be a transmission source.

In addition, the potential risk of transmission via feces had implications on fecal microbiota transplantation (FMT) highly effective for recurrent *Clostridium difficile* infections. It is speculated that COVID-19 might be transmitted via FMT particularly from asymptomatic donors, specifically those who tested negative for the presence of the virus in their respiratory tract but potentially positive in their fecal samples [133]. No cases of COVID-19 transmission through FMT treatment have been reported, but only FMT products generated from stool donated before December 2019 or before November 2019 can be used according to the FDA recommendations and Hong Kong guidelines, respectively.

Finally, fecal shedding may have important epidemiological implications for community surveillance tools such as wastewater monitoring, which inform public health measures. Detection of SARS-CoV-2 RNA in untreated wastewater has been reported [134]. Detecting SARS-CoV-2 in wastewater might represent a way to better surveille the status of the population and detect peaks of infection and admissions to hospital up to one week ahead development of symptoms or detection in nasopharyngeal swabs [135]. In a recent study [136] fecal viral RNA was observed up to 7 months post infection in patients with mild to moderate COVID-19. Understanding the temporal dynamics of fecal shedding in individuals with mild or even asymptomatic disease is essential for inferring population-prevalence of COVID-19 from wastewater studies [137]. Currently the majority of the longitudinal studies of fecal viral RNA shedding have been limited to hospitalized patients with severe COVID-19 and/or with co-morbidities [138]. As stated by the authors, the continued presence of fecal viral RNA in wastewater may be mistakenly interpreted as evidence of the prevalence of infectious individuals in a community. Since wastewater viral RNA levels are being considered for use in guiding community level policies, it is critical to better understand how aerosol transmissibility of SARS-CoV-2 RNA are temporally related to fecal viral RNA shedding [136]. 

More evidence is required to demonstrate whether and in which conditions SARS-CoV-2 can establish a fecal–oral transmission route. This requires determining which people are susceptible to GI infection, and from this pool, in what conditions may people shed infectious virus particles in feces. Another important outstanding question to resolve is determining the minimum infectious dose of SARS-CoV-2, which may vary for the different SARS-CoV-2 variants.

### 5.3. Gut Implication in the Severity of the COVID-19 Outcomes

While a first Asian analysis at the beginning of the pandemic suggested that the presence of GI symptoms in COVID-19 patients was associated with increased clinical deterioration [139], other European or American studies have subsequently found the opposite [140,141,142]. An initial study showed no significant correlation consistent with GI tract symptomatology and disease severity [143]. Later on, a first meta-analysis did not show a statistically significant difference in mortality between patients with or without GI symptoms [144]. A second recently published (March 2022) large meta-analysis including 53 studies with 55,245 patients also showed no association [144], but two studies including more groups and people associated GI symptoms with worse prognosis of the disease [5,145]. Thus, it is still unclear whether GI symptoms could be predictive of disease severity. 

An important body of evidence, however, supports the crucial implication of the gut in the excessive inflammatory response in COVID-19. Under normal conditions, inflammation is a protective process that combats infection. However, prolonged inflammatory response has long been known to play a detrimental role in human diseases, and clinical markers of excessive systemic inflammatory response were associated with severe and fatal COVID-19 [146,147]. Hyperinflammation contributes to broad tissue damage, acute respiratory distress syndrome, multiple-organ failure and ultimately death [148] and has been described as central in inducing severe outcomes in COVID-19 patients [149,150]. Impaired intestinal barrier function enhances the translocation of gut bacteria and of bacterial toxins, such as peptidoglycans and lipopolysaccharides (LPS), from the gut lumen into the blood. Increased levels of LPS in the blood (endotoxemia) activate Toll-Like Receptors, leading to the production of numerous pro-inflammatory cytokines and, hence, low-grade systemic inflammation [151]. In severely ill patients, intestinal barrier disruption and associated bacterial translocation exacerbates systemic inflammation [152,153]. Three studies found higher gut permeability markers in (severe) COVID-19 patients with abnormal presence of gut microbes in their bloodstream [7,91,92]. High levels of zonulin (gut permeability marker) were associated with severe COVID-19 and bacterial products in the blood correlated strongly with higher levels of markers of systemic inflammation and immune activation (such as C-reactive peptide levels) [7,91,92,123]. This does not imply that microbial translocation is the primary trigger of the inflammation, but supports the hypothesis that disrupted intestinal barrier and associated bacterial translocation play an additive or synergistic role in the cytokine storm underlying severe COVID-19 [92,154]. In addition, bacteria translocation from the gut into the systemic circulation might result in secondary infections and aggravate pulmonary symptoms in COVID-19 patients [155,156]. 

Disruption of the intestinal barrier also induces a local inflammatory response. It remains unclear whether permeability changes are primary events or secondary effects triggered by inflammation. Increased intestinal permeability and chronic intestinal inflammation are hallmarks of inflammatory bowel diseases (IBD), such as Crohn’s disease (CD) [157]. Taking advantage of the genetic aspect in CD, several studies reported that increased permeability might precede CD onset as abnormal lactulose-to-mannitol ratios in asymptomatic first-degree relatives of CD patients was associated with a CD diagnosis during the follow up time [158,159,160]. In line, in the IL-10 gene-deficient IBD mouse model, increased intestinal permeability was observed early in life and then mice spontaneously developed colitis at 12 weeks age [161]. In addition, IL-10 deficient animals treated with AT-1001, a zonulin peptide inhibitor previously shown to reduce small intestinal permeability, developed less colitis later in life. Results from IBD mouse models suggest that investigation of intestinal permeability and inflammation in SARS-CoV-2 infected mice or cells treated with AT-1001 could be informative of the sequential process. Recently, a drug repurposing approach identified AT-1001, currently in phase 3 trials in celiac disease, as a potential therapeutic approach for COVID-19, however still requiring optimization steps [162]. In light of the central role of inflammation in COVID-19, concerns were raised that IBD patients may have an increased risk of worse outcomes. Corticosteroids, commonly used medications for IBD, were associated with adverse outcomes in COVID-19, but overall IBD patients did not have an increased risk of COVID-19 and had largely similar outcomes as the general population [163]. 

Finally yet importantly, associations between levels of inflammatory markers and gut microbiota composition in COVID-19 patients suggest that the gut microbiota might be involved in the magnitude of COVID-19 severity [122]. Significant alterations in fecal microbiomes of COVID-19 patients were reported at all times of hospitalization [122,164,165,166]. Recently, animal studies in mice, hamsters and nonhuman primates provided evidence that SARS-CoV-2 infection directly alters the gut microbiome [6,87,166]. However, the underlying mechanisms are still poorly understood. A body of evidence supports that intestinal and systemic inflammation, dysregulation of intestinal ACE2 or infection of intestinal bacteria can be interconnected pathways leading to gut dysbiosis as an adverse outcome following SARS-CoV-2 in the gut, but further laboratory research and large-scale population-based studies are needed to validate these pathways [167]. In addition, changes in the lung microbiome with increase of bacteria normally found in the GI tract were reported in COVID-19 patients [168]. Besides, gut dysbiosis during respiratory viral infection has been shown to worsen lung pathology and to promote secondary infections [156]. Based on the current insights, modulating the gut microbiota with probiotics, prebiotics or diet to improve disease prevention and management might represent easy to implement strategies [169]. Clinical trials in COVID-19 of probiotics with expected anti-inflammatory effects in the gut–lung axis are currently underway [170]. 

### 5.4. Gut Implication in Long COVID

Finally, GI disorders described in patients appeared to precede, accompany or follow the respiratory symptoms [5,171,172]. Long-term sequelae of COVID-19, collectively termed the post-acute COVID-19 syndrome (PACS) or long COVID, are rapidly emerging across the globe and many studies following patients who have recovered from the respiratory effects of COVID-19 identified persistent GI sequelae [173,174,175]. In a study from China, around half of the patients (41 of 74) had fecal samples positive for SARS-CoV-2 RNA, which remained positive for longer than the respiratory samples [173]. A recent study detected fecal RNA in around half of participants (113 patients with mild to moderate COVID-19) within the first week after diagnosis and around 4% of the patients shed up viral RNA up to 7 months after diagnosis while respiratory samples were negative [136]. No association between symptomatology and fecal viral RNA shedding was found in this study in participants with active respiratory infection, but when focusing on participants with extended shedding of fecal viral RNA after respiratory shedding ceased, fecal viral RNA was associated with GI symptoms [136]. Another recent study using a cohort of IBD patients showed that SARS-CoV-2 antigens could persist in the gut up to 7 months after infection. Importantly, only those IBD with detectable viral RNA in the gut were found to display post-acute COVID-19 symptoms [176]. In non-human primates infected with SARS-CoV-2, the viral RNA load decreased less rapidly over time in rectal samples than in nasopharyngeal and tracheal swabs [6]. These studies support the possibility that a prolonged SARS-CoV-2 presence in the GI tract, after the respiratory infection is cleared, might represent long-term viral reservoirs contributing to long COVID. A potential bacteriophage-like behavior of SARS-CoV-2 might also offer a way to explain the intestinal/fecal long-term presence of SARS-CoV-2. While the pathogenesis of long COVID is still under intense investigation, on the four current leading hypotheses [63], it is interestingly to note that two involve the gut: (i) gut dysbiosis [177,178] and (ii) viral reservoir with residual SARS-CoV-2 viral antigens [179] and persistent SARS-CoV-2 nucleic acids [76,177] reported in GI tissues in patients months after diagnosis and proposed to drive chronic inflammation. However, the concept that viral antigen persistence instigates immune perturbation and post-acute COVID-19 still requires validation in controlled clinical trials [176]. Towards that end, the RECOVER initiative (https://recovercovid.org/about) aims to bring together patients, caregivers, clinicians and scientists to understand, prevent and treat Long COVID, notably by collecting biopsies from the lower intestines of some participants [180]. Continuing the unprecedented degree of scientific collaboration, such unified interdisciplinary actions to collect and characterize sufficient PASC cases will enable to identify which factors affects long COVID. In addition, animal models such as humanized mice [181] or Syrian hamsters [182] will help to highlight molecular mechanism of long COVID and to explore future therapeutics. 

Finally, currently the definition of long COVID differs depending of the health organizations [183,184]. There is a need for either a universal definition or to stop treating long COVID as a single entity as this umbrella term might represent multiple conditions (203 symptoms reported in 10 organs systems) [185]. Defining long COVID (categories) will help deciphering underlying mechanisms to ultimately improve disease prevention, management and treatment.

## 6. Conclusions

There are multiple outstanding questions regarding SARS-CoV-2 interaction with the human gut. First, it is not firmly established whether SARS-CoV-2 can actively replicate in human intestine. Evidence from multiple in vitro and in vivo animal studies points towards a direct viral tropism of intestinal cells and a productive enteric infection by SARS-CoV-2, however species, dose, virus preparations and route of inoculum are important factors to consider that can influence the occurrence of productive intestinal infection in animal studies. In addition, it is possible that specific conditions increase susceptibility to SARS-CoV-2 replication in the gut. Further studies are clearly needed to determine the experimental and clinical conditions under the gut represents an alternative entry route for the virus into the body. Such conditions encompass comorbidities, age, medication, inflammatory status, dysbiosis, fasted-fed status or ingestion with food. Secondly, based on the current evidence, it remains unclear whether GI symptoms, and particularly diarrhea, are caused by direct infection of the GI tract by SARS-CoV-2 or whether they are a consequence of a local and systemic immune activation. The wide range in reported rates of diarrhea in clinical studies of SARS-CoV-2 positive patients (from as low as 2% up to 50%) calls for more clinical studies and meta-analysis to elucidate the percentage of COVID-19 patients who develop GI symptoms, and particularly diarrhea, and whether GI disorders depend on active SARS-CoV-2 enteric infection and/or on factors such as those cited above. Answering those questions will be important for deciding the course of medical treatment. Thirdly, at this time, there is a moderate level of evidence to support the idea that the GI tract serves as an alternative route of virus dissemination. Finally, the potential implication of the gut on long COVID possibly by acting as viral reservoir or due to alteration of gut microbiota requires and deserves significant further investment in research, treatment and care of the PACS patients. In conclusion, in addition to calling for further research and large-scale studies, the potential impacts of SARS-CoV-2 productive enteric infection recommends applying appropriate precautions and potential preventive actions. 

## Figures and Tables

**Figure 1 jcm-11-05691-f001:**
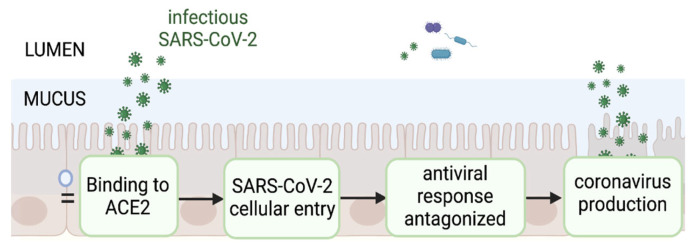
Pathway depicting the proposed sequence of events of a productive SARS-CoV-2 infection in the gut. Virus binding to the ACE2 receptor expressed on enterocytes mediates viral entry inducing an antiviral response that must be antagonized for new virions to be produced. This manuscript will evaluate available published data on the likelihood of their occurrence.

**Figure 2 jcm-11-05691-f002:**
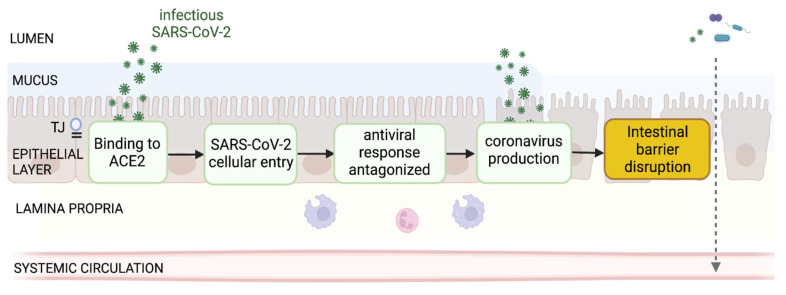
Pathway depicting the sequence of events for SARS-CoV-2 enteric production impairing intestinal barrier (AOP422). Evidence is evaluated in the present work to assess the likelihood of the occurrence.

**Figure 3 jcm-11-05691-f003:**
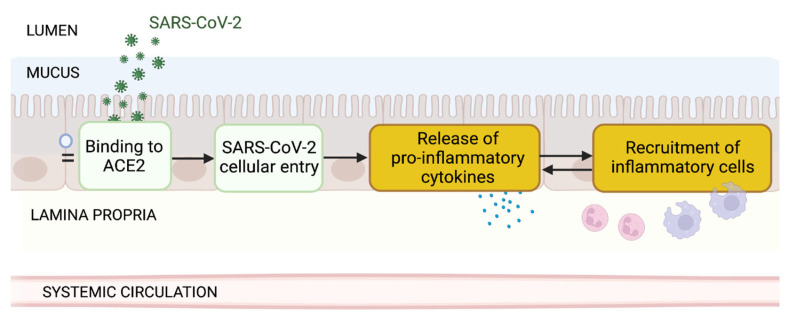
Pathway depicting the sequence of events of SARS-CoV-2 enteric infection contributing to the inflammatory response. This manuscript assesses whether evidence supports these events.

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
