# Peer review of "Gut as an Alternative Entry Route for SARS-CoV-2: Current Evidence and Uncertainties of Productive Enteric Infection in COVID-19"

_jcm, 2022, doi:10.3390/jcm11195691_

Round 1

Reviewer 1 Report

This review deals with the impact of a SARS-CoV-2 infection on gut with a strong focus on virus replication. This is timely and relevant. The review is particularly well writen and presents some novelties. The review is very well balanced (e.g. arguments and counter-arguments). It synthetizes the litterature very well although some points probably need to be improved. Below are the main issues.

It is surprising that the impact of infection on gut microbiota’s composition is not detailed. The review requires a chapter describing the major changes both in humans and animal models (page 10). It does not need to be long as this is not the main focus of the review but I think it would add information regarding the potential consequences of virus/gut interactions. This is all the more important since the authors mentioned the impact of the infection on gut barrier properties (well known to be influenced by the gut microbiota).

I am surprised by this sentence (page 6) : « In patients, several studies indicate that if IFN is administered intranasally just before or upon exposure, SARS-CoV-2 viral tropism to distal tissues, including the intestine, is limited [62] ». Did patients received IFN before infection? Did the authors collected intestine samples from these patients? The hamster system was used here.

Page 8 :

« In COVID-19 patients, a small study showed that fecal calprotectin… «  

What do you mean by « a small study »?

« In another exploratory study, COVID-19 patients had elevated plasma levels of LPS-binding protein (a gut leakage marker) but not of intestinal FABP (a marker of enterocyte damage) [88]» .

iFABP has also been detected in SARS-CoV-2-infected hamsters (there is a trend at least, refs 84).

« Severe COVID-19 was associated with high levels of markers of tight junction permeability and microbial translocation [1], [89], [90], signaling a loss of the intestinal barrier functio » plus page 12.

Any clues in experimental models ?

Page 10

« Individuals with altered intestinal barrier prior to infection, or under certain medication or comorbidities, might be more vulnerable to gastrointestinal SARS-CoV-2 infection [112]. »

Very true. Any experimental models in progress to investigate this important question?

« Gut dysbiosis which has been associated with severity in COVID-19 [114], [115], [116] might contribute… »

Original papers, and not reviews (those one are not the best), should be quoted here.

« bacteriophage-like behavior of SARS-CoV-2 ».

 I would be much more cautious. Ref 118 is a bit «vague».

Page 13 « Long COVID ».

Do experimental models mimic long covid? The authors might want to quote this excellent paper : Frere, J. J. et al. SARS-CoV-2 infection in hamsters and humans results in lasting and unique systemic perturbations post recovery. Science Translational Medicine 0, eabq3059 (2022).

Author Response

Response to Reviewer 1 Comments

This review deals with the impact of a SARS-CoV-2 infection on gut with a strong focus on virus replication. This is timely and relevant. The review is particularly well writen and presents some novelties. The review is very well balanced (e.g. arguments and counter-arguments). It synthetizes the literature very well although some points probably need to be improved. Below are the main issues.

We thank the reviewer for his/her comment that we have addressed point by point here below.

Point 1. It is surprising that the impact of infection on gut microbiota’s composition is not detailed. The review requires a chapter describing the major changes both in humans and animal models (page 10). It does not need to be long as this is not the main focus of the review but I think it would add information regarding the potential consequences of virus/gut interactions. This is all the more important since the authors mentioned the impact of the infection on gut barrier properties (well known to be influenced by the gut microbiota).

Response: We totally agree with the reviewer that the impact on gut microbiota is important to discuss in this context. We actually describe this aspect in details in another paper submitted in parallel to this Special Issue. To make the message clear for the readers, we focused on current evidence related to intestinal virus replication in this review and on the evidence regarding gut microbiota alteration in the other. We asked the editor how to cite this parallel review submitted in this Special Issue.

However, we agree that the two aspects are interconnected along with the gut barrier proprieties and intestinal inflammation. To cover the impact of direct infection on gut dysbiosis, we added a paragraph in this revised version section 5.3 p.15

Finally yet importantly, associations between levels of inflammatory markers and gut microbiota composition in COVID-19 patients suggest that the gut microbiota might be involved in the magnitude of COVID-19 severity (10.1136/gutjnl-2020-323020). Significant alterations in fecal microbiomes of COVID-19 patients were reported at all times of hospitalization (10.1093/cid/ciaa709; 10.1136/gutjnl-2020-323020; 10.1053/j.gastro.2020.05.048; 10.21203/rs.3.rs-726620/v1). Recently, animal studies in mice, hamsters and nonhuman primates provided evidence that SARS-CoV-2 infection directly alters the gut microbiome (12; 84; 10.21203/rs.3.rs-726620/v1). However, the underlying mechanisms are still poorly understood. A body of evidence supports that intestinal and systemic inflammation, dysregulation of intestinal ACE2 or infection of intestinal bacteria can be interconnected pathways leading to gut dysbiosis as an adverse outcome following SARS-CoV-2 in the gut, but further laboratory research and large-scale population-based studies are needed to validate these pathways (Clerbaux et al, 2022b). In addition, gut dysbiosis during respiratory viral infection has been shown to worsen lung pathology and to promote secondary infections (10.1016/j.jiph.2020.07.003). Based on the current insights, modulating the gut microbiota with probiotics, prebiotics or diet to improve disease prevention and management might represent easy to implement strategies (10.2760/54454). Clinical trials in COVID-19 of probiotics with expected anti-inflammatory effects in the gut–lung axis are currently underway (10.1016/j.nmni.2021.100837).”  

In addition gut dysbiosis, prior to infection, might be a risk factor for severe COVID-19. This is mentioned p.12 of the revised version “A body of evidence indicates that gut dysbiosis, prior to infection, represents a risk factor, meaning contributes to more severe outcomes in COVID-19 patients, potentially by modulating intestinal ACE2 expression, intestinal and systemic inflammation and gut barrier integrity (10.3390/jcm11154464)”.  

Point 2. I am surprised by this sentence (page 6) : « In patients, several studies indicate that if IFN is administered intranasally just before or upon exposure, SARS-CoV-2 viral tropism to distal tissues, including the intestine, is limited [62] ». Did patients received IFN before infection? Did the authors collected intestine samples from these patients? The hamster system was used here.

Response: We agree that the writing of the sentence was not correct as such, we thank the reviewer for this comment. The sentence has been modified p.6 of the revised version “Studies in human primary nasal epithelial cell cultures have shown that if exogenous IFN-I/III were administered intranasally prior to infection and at sufficient concentration, SARS-CoV-2 infection was inhibited [79]. Furthermore, in a hamster model IFN treatment limited tropism to distal tissues, including the intestine [62]” with the reference Hatton [79] for patients and Hoaglund [62] for hamster.

Point 3. Page 8 : « In COVID-19 patients, a small study showed that fecal calprotectin… «  What do you mean by « a small study »?

Response: Effenberger et al. analyzed 40 hospitalized patients with COVID-19 at the University Hospital of Innsbruck, Austria. We modified the sentence in the revised version as such p.9 “In humans, no relation was noted between fecal calprotectin (FC) levels and fecal SARS-CoV-2 RNA in a cohort of 40 hospitalized patients with COVID-19”.

Point 4. « In another exploratory study, COVID-19 patients had elevated plasma levels of LPS-binding protein (a gut leakage marker) but not of intestinal FABP (a marker of enterocyte damage) [88]» . iFABP has also been detected in SARS-CoV-2-infected hamsters (there is a trend at least, refs 84).

Response: We thank the reviewer for having brought our attention to this interesting data. We added this information in the revised version p.9 “In Syrian hamsters, the histological analysis did not reveal intestinal damage or structural remodeling of the epithelium in hamsters but a trend towards increased blood concentration of intestinal fatty-acid binding protein (FABP), a systemic marker associated with disrupted gut integrity, has been detected [84].”

Point 5. « Severe COVID-19 was associated with high levels of markers of tight junction permeability and microbial translocation [1], [89], [90], signaling a loss of the intestinal barrier functio » plus page 12. Any clues in experimental models ?

Response: To our knowledge, this parameter has been not been measured in the experimental models described above and for which other intestinal analysis have been performed (mainly changes in the microbiota composition following SARS-CoV-2 infection https://doi.org/10.3389/fcimb.2021.792584).

We agree that it would be definitively informative to assess those aspects in these models. We added a sentence in the gaps paragraph in the revised manuscript p.9 “In addition, it would informative to assess the tight junction permeability in infected mice, hamsters or nonhuman primate models described above.”

Point 6. Page 10 « Individuals with altered intestinal barrier prior to infection, or under certain medication or comorbidities, might be more vulnerable to gastrointestinal SARS-CoV-2 infection [112]. »

Very true. Any experimental models in progress to investigate this important question?

Response: Different experimental models mimicking diseases known to be associated with an altered intestinal barrier exist. Literature describing their use to unravel the mechanisms behind SARS-CoV-2 GI infection is starting to emerge. A preclinical T2DM/obesity co-morbidity model of COVID-19 was established in mice (10.3389/fcimb.2021.792584). A mouse model able to mimic obesity-associated COVID-19 comorbidities using a mouse-adapted SARS-CoV-2 strain could help elucidate pathogenesis that may be occurring in humans with obesity (10.3389/fcimb.2021.792584). Such models could accelerate the development of therapeutics for this highly susceptible population. Sex and diet-specific responses beginning to explain the effects of obesity and diabetes on COVID-19 disease were observed (10.3389/fcimb.2021.792584). The detrimental impact of continuous Western diet on COVID-19 outcome has been proved in Syrian hamsters (10.1101/2021.06.17.448814). The age dependent increase in disease can be observed in Syrian hamsters and nonhuman primates (87). Thus age, medication, metabolic syndrome, via high fat diet for example, should be incorporated into models to mimic human comorbidities in order to investigate this important question.

The text has been modified (p12) to generate awareness regarding the emergence of this type of study.

Point 7. « Gut dysbiosis which has been associated with severity in COVID-19 [114], [115], [116] might contribute… » Original papers, and not reviews (those one are not the best), should be quoted here.

Response: We changed the references to original papers (10.1136/gutjnl-2020-323020; 10.3389/fmicb.2021.705020).

Point 8. « bacteriophage-like behavior of SARS-CoV-2 ». I would be much more cautious. Ref 118 is a bit «vague».

Response: We modified the paragraph to meet the comment page 12 of the revised version. “A recent study showed that SARS-CoV-2 replicates outside the human body in vitro in bacterial growth medium, following bacterial growth and influenced by antibiotics administration, suggesting a bacteriophage-like behavior for SARS-CoV-2 [117] or the activation of other bacteriophages [119]. Electron and fluorescence microscopy images showed the presence of SARS-CoV-2 both outside and inside bacteria [118], [119]. These observations, compatible with a bacteriophage-like behavior of SARS-CoV-2, could propose a new lens for understanding COVID-19 transmission.”

Point 9. Page 13 « Long COVID ». Do experimental models mimic long covid? The authors might want to quote this excellent paper :         Frere, J. J. et al. SARS-CoV-2 infection in hamsters and humans results in lasting and unique systemic perturbations post recovery. Science Translational Medicine 0, eabq3059 (2022).

Response: We thank the reviewer for sharing this interesting paper. We added a sentence with the reference mentioned In addition, animal models such as humanized mice (10.1038/s41587-021-01155-4) or Syrian hamsters (10.1126/scitranslmed.abq3059) will help to highlight molecular mechanism of long COVID and to explore future therapeutics.

Reviewer 2 Report

1.       Authors described the gut-lung axis and its possible role in COVID-19, however gut-lung dysbiosis during COVID-19 also enhances the chances of secondary infections such as diarrhea. Please follow the references and discuss.

https://pubmed.ncbi.nlm.nih.gov/33425362/

https://www.ncbi.nlm.nih.gov/pmc/articles/PMC7359806/

2.       ACE2 receptor plays an essential role during the SARS-CoV-2 entry, and ACE2 is also highly expressed in the gut. This suggests the possible entry of SARS-CoV-2 via the enteric ACE2. However, NRP1 is another potential receptor that is suggested to assist the cellular entry of SARS-CoV-2 during the infection. Interestingly, NRP1 is also highly expressed in the gut. Please discuss the possibilities and follow the below reference.

https://pubmed.ncbi.nlm.nih.gov/35447881/

3.       Overall, lung and gut inflammation plays a significant role in COVID-19. There is an opportunity to make a model figure, showing the link between gut-lung dysbiosis and associated inflammation during COVID-19. Reviewers suggested including a summarized model figure to make this review more interactive.

4.       Please also discuss the impact of vaccination on long COVID-19 and associated management strategies. Possibly a separate section for long COVID management strategies in the light of facts provided in the present review. 

Author Response

Response to Reviewer 2 Comments

We thank the reviewer for his/her comment that we have addressed point by point here below.

Point 1. Authors described the gut-lung axis and its possible role in COVID-19, however gut-lung dysbiosis during COVID-19 also enhances the chances of secondary infections such as diarrhea. Please follow the references and discuss.https://pubmed.ncbi.nlm.nih.gov/33425362/

https://www.ncbi.nlm.nih.gov/pmc/articles/PMC7359806/

Response: We added a paragraph in section 5.3 p.15 of the revised manuscript pointing to the references mentioned here. A body of evidence supports that intestinal and systemic inflammation, dysregulation of intestinal ACE2 or infection of intestinal bacteria can be interconnected pathways leading to gut dysbiosis as an adverse outcome following SARS-CoV-2 in the gut (Clerbaux et al, 2022b), but further laboratory research and large-scale population-based studies are needed to validate these pathways. In addition, changes in the lung microbiome with increase of bacteria normally found in the GI tract were identified in COVID-19 patients (10.3389/fmicb.2020.01302). Besides, gut dysbiosis during respiratory viral infection has been shown to worsen lung pathology and to promote secondary infections (10.1016/j.jiph.2020.07.003). Based on the current insights, modulating the gut microbiota with probiotics, prebiotics or diet to improve disease prevention and management might represent easy to implement strategies (10.2760/54454). Clinical trials in COVID-19 of probiotics with expected anti-inflammatory effects in the gut–lung axis are currently underway (10.1016/j.nmni.2021.100837). 

Point 2. ACE2 receptor plays an essential role during the SARS-CoV-2 entry, and ACE2 is also highly expressed in the gut. This suggests the possible entry of SARS-CoV-2 via the enteric ACE2. However, NRP1 is another potential receptor that is suggested to assist the cellular entry of SARS-CoV-2 during the infection. Interestingly, NRP1 is also highly expressed in the gut. Please discuss the possibilities and follow the below reference.https://pubmed.ncbi.nlm.nih.gov/35447881/

Response: We had not included neuropilin-1 (NRP-1) before because its role in the gut has not been investigated specifically, but we agree with the reviewer that it makes sense to discuss this receptor and we thank him/her for pointing this out. We modified section 2 to accommodate current knowledge on NRP-1.

p.3 in the revised version in the Biological plausibility section: In addition, neuropilin-1 (NRP-1) was proposed to act as ACE2 co-receptor and promote, although to very low levels, SARS-CoV-2 entry even in cells that lack ACE2 and TMPRSS2 (10.1126/science.abd2985). Maximum infection was reported when NRP-1 and ACE2 are co-expressed on the same cell types (10.1126/science.abd3072). NRP-1 is reported to be expressed in the epithelia of the GI tract (10.1126/science.aal3321´).”

p.5 in the Uncertainties and gaps section: No studies have specifically investigated the role of NRP-1 in SARS-CoV-2 entry in the gut. However, it is interesting to note that different variants display different affinities for NRP-1, with omicron displaying higher affinity than previous variants. Future studies should elucidate whether this increase in affinity constitutes a functional evolutionary adaptation of SARS-CoV-2 to humans (10.3390/idr14020029), and confer an advantage for viral entry.”

Point 3.  Overall, lung and gut inflammation plays a significant role in COVID-19. There is an opportunity to make a model figure, showing the link between gut-lung dysbiosis and associated inflammation during COVID-19. Reviewers suggested including a summarized model figure to make this review more interactive.

Response: We agree that there is an interesting interconnection between the gut and the lung in this disease, including at the level of the microbiota alteration. Intestinal implication is broad in COVID-19, gut-liver or gut-brain axis are also of great interest. However, we aim in this review to focus on the gut only and on the evidence currently available in the literature regarding the possibility of the gut being an alternative entry route for the virus into the body. We explored evidence for each of the figures we proposed in the manuscript. A model figure will be more hypothetical and will require a new review of the literature on that topic.

We added two paragraphs to discuss these aspects in the section 5.3 of the revised version:

p.14In addition, bacteria translocation from the gut into the systemic circulation might result in secondary infections and aggravate lung symptoms in COVID-19 patients (10.1101/2021.07.15.452246; 10.1016/j.jiph.2020.07.003).

p.15In addition, changes in the lung microbiome with increase of bacteria normally found in the GI tract were identified in COVID-19 patients (10.3389/fmicb.2020.01302). Besides, gut dysbiosis during respiratory viral infection has been shown to worsen lung pathology and to promote secondary infections (10.1016/j.jiph.2020.07.003). Based on the current insights, modulating the gut microbiota with probiotics, prebiotics or diet to improve disease prevention and management might represent easy to implement strategies (10.2760/54454). Clinical trials in COVID-19 of probiotics with expected anti-inflammatory effects in the gut–lung axis are currently underway (10.1016/j.nmni.2021.100837). 

Point 4. Please also discuss the impact of vaccination on long COVID-19 and associated management strategies. Possibly a separate section for long COVID management strategies in the light of facts provided in the present review. 

Response: The impact of vaccination on long COVID is definitively an important question. Currently, the consensus among the scientific community is that we do not know what long COVID is (no consensus on the definition) and we do not have enough well characterized cases, notably to be able to evaluate the impact of vaccination on long COVID.

We invite the reviewer to watch the replies to this crucial question of two renowned experts of long COVID, Dr Putrino and Dr Iwasaki (min 49:01 – https://youtu.be/sJiUePEDrWg; https://www.youtube.com/watch?v=sJiUePEDrWg).

We added a paragraph in section 5.4 on long COVID to discuss management strategies in the revised manuscript p.16

“While the pathogenesis of long COVID is still under intense investigation, on the four current leading hypotheses (10.1126/science.abm8108), it is interestingly to note that two involve the gut: (i) gut dysbiosis (10.1097/PG9.0000000000000152; 10.1136/gutjnl-2021-325989) and (ii) viral reservoir with residual SARS-CoV-2 viral antigens (10.1136/gutjnl-2021-324280) and persistent SARS-CoV-2 nucleic acids (10.1038/s41586-021-03207-w; 10.1097/PG9.0000000000000152) reported in GI tissues in patients months after diagnosis and proposed to drive chronic inflammation. However, the concept that viral antigen persistence instigates immune perturbation and post-acute COVID-19 still requires validation in controlled clinical trials [154]. Towards that end, the RECOVER initiative (https://recovercovid.org/about) aims to bring together patients, caregivers, clinicians and scientists to understand, prevent and treat Long COVID, notably by collecting biopsies from the lower intestines of some participants [155]. Continuing the unprecedented degree of scientific collaboration, such unified interdisciplinary actions to collect and characterize sufficient PASC cases will enable to identify which factors affects long COVID. In addition, animal models such as humanized mice (10.1038/s41587-021-01155-4) or Syrian hamsters (10.1126/scitranslmed.abq3059) will help to highlight molecular mechanism of long COVID and to explore future therapeutics. Finally, currently the definition of long COVID differs depending of the health organizations (https://www.who.int/news-room/questions-and-answers/item/coronavirus-disease-(covid-19)-post-covid-19-condition; https://www.cdc.gov/coronavirus/2019-ncov/long-term-effects/index.html). There is a need for either a universal definition or to stop treating long COVID as a single entity as this umbrella term might represent multiple conditions (203 symptoms reported in 10 organs systems) (10.1016/j.eclinm.2021.101019). Defining long COVID (categories) will help deciphering underlying mechanisms to ultimately improve disease prevention, management and treatment.

Reviewer 3 Report

Authors have done a literature review of current evidence on enteric infection of SARS-CoV-2. They have also included animal and human studies. There are some key points (in details below) which needs to be significantly improved before any submission. 

First this is a broad topic and many other factors such as dose of inoculum or virus might affect on productive infection which authors have not well discussed. Second the objective is not very clear if authors are trying to establish and introduce their methodological approach to the readers. Third there is no method, result and discussion for this manuscript, Only abstract, introduction, and conclusions. Fourth, the title of sections are not consistent with the contents of sections which needs to be revised.   

Introduction

1)    Please provide the reference for “The strength of the relationship between the events is established by demonstrating biological plausibility and causality between pairs of events, called key event relationships (KER)”.

2)    Page 4, the sentence is not finished “in lungs than in”.

3)    In Ref 37, It’s not clarified whether lethality has been observed by IG route or other routes “Another transgenic mouse model with hACE2 driven by the heterologous promoters (K18-hACE2) showed lethal disease”.

4)     It would be helpful if authors clarified “that study” exactly refers to which study. after reading article in ref, I assume no level observed in the gut instead of “no high levels”.

5)    Authors are categorized the animal studies (i.e., hAce2 mice, hamster,) as a subdivision or under the title “2.1. S proteins bind to ACE2 in enterocytes mediates viral entry, Evidence” that sounds inappropriate and inconsistent to the title, because they are reviewing the current studies on observing the infection by GI tract.

6)    Similar as mentioned above is also for mentioned human study in ref 41. Moreover, it would be helpful if authors clarified here “the virus could productively replicate, but not as well as SARS-Co-V” the virus exactly refers to which virus.

7)    Please provide the reference for these two sentences “In all animal studies, viral RNA was detected in the intestines after virus inoculation (intranasal or intragastric), providing strong evidence for SARS-CoV-2 entry into enterocytes”.

and, for “However, replicative virus was noticed only in the gut of rhesus monkeys”.

8)    Please provide the reference for “In addition, these data might also suggest that the viral RNA detected was residual input inoculum from the nasal mucosa transferred to the intestines by swallowing or that an antiviral response blocks viral replication”.

Does it refer to animal studies? Please clarify more the correlation of antiviral response blocks viral replication with viral RNA detected.

9)    Please provide the reference “In humans, according to the few endoscopic and histological examinations based on one or two cases, the GI epithelium is potentially susceptible to infection by SARS-CoV-2”.

10) Please provide the reference “Importantly to infect intestinal cells, viable SARS-CoV-2 must reach the gut lumen as an infectious particle” It would be helpful if authors clarified the infectious particle of SARS-CoV-2 with a bit more information.

11) Please provide the references for the first eight sentences of “Biological plausibility”

Conclusion

As authors have discussed for productive gut infection, the virus needs to be viable, in a large amount in gut to cause productive infection in gut. As I assume there are other important factors such as dose and route of inoculum in animal studies that cause or may result to productive gut infection compared to other routes. None of the above factors have not been discussed in mentioned animal studies. I question authors how they conclude productive enteric infection based on animal studies or they may have considered only the presence of Viral RNA in gut (in that case I assume it was also not proved and observed in all mentioned studies)

Author Response

Response to Reviewer 3 Comments

Authors have done a literature review of current evidence on enteric infection of SARS-CoV-2. They have also included animal and human studies. There are some key points (in details below) which needs to be significantly improved before any submission. 

First this is a broad topic and many other factors such as dose of inoculum or virus might affect on productive infection which authors have not well discussed. Second the objective is not very clear if authors are trying to establish and introduce their methodological approach to the readers. Third there is no method, result and discussion for this manuscript, Only abstract, introduction, and conclusions. Fourth, the title of sections are not consistent with the contents of sections which needs to be revised.   

Response: We agree with the reviewer that the dose and the route of inoculum in animal studies are important factors to discuss. We briefly mentioned those points in the review in section 2.3 but we added additional paragraphs about dose of inoculum to four sections:

Section 2.3 Coronavirus production

p.7 “Thus, multiple strategies for introducing hACE2 into mice have been developed, a comprehensive characterization of the different models as well as of the doses and routes of inoculum used in each case is needed to correctly interpret the results [87].

p.7 “As already mentioned, these data calls for precaution of which models are suitable to study SARS-CoV-2 intestinal infection as well as for considering with care the doses and routes of inoculum used in each case”

Section 5.1 Productive enteric infection

p.12 In infected hamsters, SARS-CoV-2 intranasal infection was more efficient than oral infection. However, increasing viral dose in the initial inoculum, both intranasal and oral, resulted in higher levels of SARS-CoV-2 RNA in the lungs and in the intestines of these animals, suggesting that the initial dose is an important factor when considering gut infection and mechanisms that protect the virus from the harsh environment of the stomach (10.1016/j.xcrm.2020.100121). Alternatively, other cell types may be able to transport SARS-CoV-2 to the gut, as for example, a small number of lymphocytes has been shown to be infected by SARS-CoV-2 (10.1093/cid/ciaa925) or even bacteria.

Section 5.2 Infectious virus in the feces

p.14 “Another important outstanding question to resolve is determining the minimum infectious dose of SARS-CoV-2, which may vary for the different SARS-CoV-2 variants.”

Conclusion

p.17 “..however species, dose, virus preparations and routes of inoculum are important factors to consider that can influence the occurrence of productive intestinal infection in animal studies…”

Regarding second and third point, there are no methods and results sections because this manuscript is a review of the literature. However, the AOP framework is a methodology currently well established in toxicology for chemical risk assessment in a regulatory context. The objective of our review here is double:

  • exploring the evidence with this approach to identify knowledge gaps and uncertainties which allows to guide further research needs and to discuss potential impact on clinical management
  • applying for the first time this methodology to the biomedical field and particularly to a viral disease – thus introducing the methodological approach to biomedical researchers and clinicians.

We added a paragraph p.2 on the context and project to clarify the objective of this review.

This study was realized under the CIAO project which aims to make sense of the overwhelming flow of publications and data related to COVID-19 pathogenesis by using the AOP framework (10.14573/altex.2102221; 10.14573/altex.2112161). The project is based on the assumption that such mechanistic organization of the COVID-19 knowledge across the different biological levels will improve the interpretation and efficient application of the scientific understanding of COVID-19 (10.3389/fpubh.2021.638605). In addition, we applied this methodology for the first time to map a viral disease of high societal relevance, expanding the AOP scope outside the toxicological field.

Fourth, we moved the content of some sections under other title sections as described in details below (point 5 and point 10).

Point 1.  Introduction. Please provide the reference for “The strength of the relationship between the events is established by demonstrating biological plausibility and causality between pairs of events, called key event relationships (KER)”.

Response: The OECD developed a guidance document outlining methods for assessing AOPs’ weight of evidence (25) and Villeneuve et al (24) describe the biological plausibility and empirical evidence principles on which KER develop. We added the references accordingly in the revised manuscript p.2

Point 2. Page 4, the sentence is not finished “in lungs than in”.

Response: We deleted the sentence.

Point 3. In Ref 37, It’s not clarified whether lethality has been observed by IG route or other routes “Another transgenic mouse model with hACE2 driven by the heterologous promoters (K18-hACE2) showed lethal disease”.

Response: In this study, the mice were inoculated via the intranasal route. We added this information to the sentence in the revised version p.7 “Another transgenic mouse model with hACE2 driven by the heterologous promoters (K18-hACE2) showed no viral RNA in the GI tract following nasal inoculation [37].”

Point 4. It would be helpful if authors clarified “that study” exactly refers to which study. after reading article in ref, I assume no level observed in the gut instead of “no high levels”.

Response: We modified the sentence Another transgenic mouse model with hACE2 driven by the heterologous promoters (K18-hACE2) showed no viral RNA in the GI tract following nasal inoculation [37], but importantly hACE2 expression was not detected in the gut in these mice [37].”

Point 5. Authors are categorized the animal studies (i.e., hAce2 mice, hamster,) as a subdivision or under the title “2.1. S proteins bind to ACE2 in enterocytes mediates viral entry, Evidence” that sounds inappropriate and inconsistent to the title, because they are reviewing the current studies on observing the infection by GI tract.

Response: We moved the evidence in the literature of a viral replication in the GI tract in different animal models following infection under the section 2.3 “coronavirus production” as this is indeed more consistent. Only the evidence of viral RNA in the gut in animal studies are still under the title reviewing evidence for viral entry into enterocytes. In addition, based on this comment of the reviewer, we added the evidence available regarding intestinal expression levels of ACE2 in animal models to be consistent with the title and objective of this section. We thank the reviewer for this comment.

Similarly, we moved the literature of viral replication in human samples to section 2.3. In addition, we added 2 studies p.6: “Another study that tested five COVID-19 patients, presenting with either upper abdominal pain or diarrhea. Early in infection, patients were subjected to a total of four esophagogastroduodenoscopy, and 2 in 5 showed signs of viral replication in the gut and increased numbers of antigen-experienced activated CD8+ T cells were detected within the epithelium (10.1038/s41385-021-00437-z). This is in line with another study that found viral nucleocapsid in 5 out of 14 patients at an average of 4 months after initial COVID-19 diagnosis (10.1038/s41586-021-03207-w).

Point 6. Similar as mentioned above is also for mentioned human study in ref 41. Moreover, it would be helpful if authors clarified here “the virus could productively replicate, but not as well as SARS-Co-V” the virus exactly refers to which virus.

Response: We revised the sentence as such “In humans, SARS-CoV-2 could productively replicate in surgically removed intestinal tissue but not in kidney or liver tissues [41]. Of note, we moved this paragraph under section 2.3 p.6 in the revised version.

Point 7. Please provide the reference for these two sentences “In all animal studies, viral RNA was detected in the intestines after virus inoculation (intranasal or intragastric), providing strong evidence for SARS-CoV-2 entry into enterocytes”. and, for “However, replicative virus was noticed only in the gut of rhesus monkeys”.

Response: We added the references to the sentence that we slightly modified p.8 in the revised version: Thus viral RNA was detected in the intestines after virus inoculation (intranasal or intragastric) in almost all animal models [35], [38], [39], [40], (10.1038/s41586-020-2324-7) providing evidence for SARS-CoV-2 entry into enterocytes. However, evidence that the virus found in the GI tissues was infectious was observed only in rhesus monkeys in one study [40].

Point 8. Please provide the reference for “In addition, these data might also suggest that the viral RNA detected was residual input inoculum from the nasal mucosa transferred to the intestines by swallowing or that an antiviral response blocks viral replication”. Does it refer to animal studies? Please clarify more the correlation of antiviral response blocks viral replication with viral RNA detected.

Response: We deleted the sentence.

Point 9. Please provide the reference “In humans, according to the few endoscopic and histological examinations based on one or two cases, the GI epithelium is potentially susceptible to infection by SARS-CoV-2”.

Response: We added the references cited above to the sentence p.8 in the revised version: “In humans, according to the few endoscopic and histological examinations based on one or two cases one or two cases ([44], [45], [46], (10.1038/s41385-021-00437-z), (10.1038/s41586-021-03207-w), the GI epithelium is potentially susceptible to infection by SARS-CoV-2”.

Point 10. Please provide the reference “Importantly to infect intestinal cells, viable SARS-CoV-2 must reach the gut lumen as an infectious particle” It would be helpful if authors clarified the infectious particle of SARS-CoV-2 with a bit more information.

Response: We moved this important aspect under section 5.1 p.11 in the revised version to be more consistent with the titles. We added a sentence to clarify the meaning of productive infection in the beginning of the section. “An important aspect is that to infect intestinal cells, viable SARS-CoV-2 must reach the gut lumen as an infectious particle, meaning able to actively replicate in the GI tract”.

Point 11. Please provide the references for the first eight sentences of “Biological plausibility”

Response: The first four sentences of section 2.2 are informed by reference 53. The citation has been added after the first sentence. This was also applied to section 2.3, where reference 65 was added after the first sentence.

Point 12. Conclusion. As authors have discussed for productive gut infection, the virus needs to be viable, in a large amount in gut to cause productive infection in gut. As I assume there are other important factors such as dose and route of inoculum in animal studies that cause or may result to productive gut infection compared to other routes. None of the above factors have not been discussed in mentioned animal studies. I question authors how they conclude productive enteric infection based on animal studies or they may have considered only the presence of Viral RNA in gut (in that case I assume it was also not proved and observed in all mentioned studies)

Response: We agree with the reviewer that the dose and the route of inoculum in animal studies are important factors to consider. As detailed above, we emphasized further the dose and route of inoculum in the revised version in section 2.3 p.7, in section 5.1 p.12, in section  5.2 p.14 and in the conclusion p.17 (see details in our first reply).

The majority of the animal studies measure the amount of viral RNA. Isolation of viable (replicative) viral particles from the gut is not straightforward for diverse reasons and despite high virus RNA concentration. As an example, Sencio et al., 2021 stated that, in their setting, they failed to detect infectious virus in the gut of Syrian hamsters (84). Similarly, only a few studies involving patients describe successful SARS-CoV-2 isolations.

Based on the evidence detailed in this review, and as highlighted in the manuscript, further studies are required to elucidate SARS-CoV-2 replication in the gut. We modified the conclusion to be more representative of the balanced arguments and counter-arguments detailed in the review and to accommodate other topics of discussion that we agree to be pertinent p.17:  “There are multiple outstanding questions regarding SARS-CoV-2 interaction with the human gut. First, it is not firmly established whether SARS-CoV-2 can actively replicate in human intestine. Evidence from multiple in vitro and in vivo animal studies points towards a direct viral tropism of intestinal cells and a productive enteric infection by SARS-CoV-2, however species, dose, virus preparations and route of inoculum are important factors to consider that can influence the occurrence of productive intestinal infection in animal studies. In addition, it is possible that specific conditions increase susceptibility to SARS-CoV-2 replication in the gut. Further studies are clearly needed to determine the experimental and clinical conditions under the gut represents an alternative entry route for the virus into the body. Such conditions encompass comorbidities, age, medication, inflammatory status, dysbiosis, fasted-fed status or ingestion with food. Secondly, based on the current evidence, it remains unclear whether GI symptoms, and particularly diarrhea, are caused by direct infection of the GI tract by SARS-CoV-2 or whether they are a consequence of a local and systemic immune activation. The wide range in reported rates of diarrhea in clinical studies of SARS-CoV-2 positive patients (from as low as 2% up to 50%) calls for more clinical studies and meta-analysis to elucidate the percentage of COVID-19 patients who develop GI symptoms, and particularly diarrhea, and whether GI disorders depend on active SARS-CoV-2 enteric infection and/or on factors such as those cited above. Answering those questions will be important for deciding the course of medical treatment. Thirdly, at this time, there is a moderate level of evidence to support the idea that the GI tract serves as an alternative route of virus dissemination. Finally, the potential implication of the gut on long COVID possibly by acting as a viral reservoir or due to gut dysbiosis requires and deserves significant further investment in research, treatment and care of the PACS patients. In conclusion, in addition to calling for further research and large-scale studies, the potential impacts of SARS-CoV-2 productive enteric infection recommends applying appropriate precautions and potential preventive actions.

Round 2

Reviewer 3 Report

Authors have not well replied to comments as requested. There are many comments but there is a few point to point responses.